# Quercetin as an Anti-Diabetic Agent in Rodents—Is It Worth Testing in Humans?

**DOI:** 10.3390/ijms26157391

**Published:** 2025-07-31

**Authors:** Tomasz Szkudelski, Katarzyna Szkudelska, Aleksandra Łangowska

**Affiliations:** 1Department of Animal Physiology, Biochemistry and Biostructure, Poznań University of Life Sciences, Wołyńska 35, 60-637 Poznań, Poland; katarzyna.szkudelska@up.poznan.pl; 2Department of Zoology, Poznań University of Life Sciences, Wojska Polskiego 71C, 60-625 Poznań, Poland; aleksandra.langowska@up.poznan.pl

**Keywords:** quercetin, diabetes, metabolism

## Abstract

Quercetin is a biologically active flavonoid compound that exerts numerous beneficial effects in humans and animals, including anti-diabetic activity. Its action has been explored in rodent models of type 1 and type 2 diabetes. It was revealed that quercetin mitigated diabetes-related hormonal and metabolic disorders and reduced oxidative and inflammatory stress. Its anti-diabetic effects were associated with advantageous changes in the relevant enzymes and signaling molecules. Quercetin positively affected, among others, superoxide dismutase, catalase, glutathione peroxidase, glucose transporter-2, glucokinase, glucose-6-phosphatase, glycogen phosphorylase, glycogen synthase, glycogen synthase kinase-3β, phosphoenolpyruvate carboxykinase, silent information regulator-1, sterol regulatory element-binding protein-1, insulin receptor substrate 1, phosphoinositide 3-kinase, and protein kinase B. The available data support the conclusion that the action of quercetin was pleiotropic since it alleviates a wide range of diabetes-related disorders. Moreover, no side effects were observed during treatment with quercetin in rodents. Given that human diabetes affects a large part of the population worldwide, the results of animal studies encourage clinical trials to evaluate the potential of quercetin as an adjunct to pharmacological therapies.

## 1. Introduction

Quercetin (3,5,7,3′,4′-pentahydroxyflavone, Figure 1) is a flavonoid compound naturally occurring in fruits and vegetables ingested by humans and animals [1,2,3]. Quercetin is one of many biologically active compounds that have attracted significant interest due to their potential therapeutic properties. Biomedical applications of biologically active compounds are of growing interest to scientists and physicians, as well as to patients and health-conscious individuals, as they offer promising possibilities for preventing and treating a wide range of diseases, typically without inducing side effects. Biologically active agents are known to exhibit properties including antioxidant, anti-inflammatory, antimicrobial, anti-aging, and anti-hypoxia. They may help prevent or treat conditions such as cancer, diabetes, asthma, neurodegenerative disorders, and cardiovascular diseases. Some compounds may also regulate the cell cycle, cell proliferation, and apoptosis. Molecular, intracellular targets through which these compounds exert therapeutic effects include the following: the suppression of pro-inflammatory cytokines, downregulation of transcription factors, and inhibition of some kinases and inducible enzymes [4,5,6,7].

The richest natural sources of quercetin include grapes, cranberries, broccoli, okra, onions, apples, cabbage, tomatoes, buckwheat, asparagus, green tea, and capers. A diet rich in quercetin-containing foods is known to exert multiple health benefits and is recommended as part of a healthy lifestyle. Owing to its broad biological activity, scientific interest in quercetin continues to grow. It has well-established antioxidant and anti-inflammatory properties [1,2,3]. Since oxidative and inflammatory stress are involved in the pathogenesis of numerous diseases [8], quercetin has been shown to mitigate a range of related conditions. These therapeutic effects include anti-aging, hepatoprotective, reno-protective, anti-obesity, neuroprotective, and immunomodulatory actions. Its cardiovascular benefits are also well documented, as it exerts antihypertensive and vasodilatory effects. Moreover, this compound can modulate cancer-associated intracellular pathways, thereby exerting anti-tumor properties. It has also been shown that quercetin acts synergistically with certain drugs and other biologically active agents, enhancing their therapeutic efficacy. Such synergistic effects have been demonstrated, for example, in combination with metformin, ampicillin, sitagliptin, epicatechin, resveratrol, curcumin, and α-tocopherol [1,2,3]. The beneficial effects of quercetin are summarized in Figure 1.

Diabetes is a global health problem, affecting approximately 10% of the world’s population. Moreover, its prevalence continues to rise. Based on etiology, symptoms, and treatment strategies, two main types of the disease are distinguished: type 1 diabetes (about 10% of cases) and type 2 diabetes (about 90% of cases). Type 1 diabetes usually has an autoimmune origin and results from the destruction of the insulin-secreting pancreatic β-cells. This leads to insulin deficiency and the resulting pronounced hyperglycemia (elevated blood glucose levels). Type 2 diabetes often develops in people who are overweight or obese. In this type of diabetes, hyperglycemia is usually moderate, but insulin resistance develops [9,10]. The progressive dysfunction of β-cells is a hallmark of both types of the disease [11,12]. If left untreated, diabetes can lead to serious complications, such as cardiovascular disease, significant vision loss, kidney dysfunction, and damage to the nervous system. One of the severe effects of diabetes is tissue necrosis, which may necessitate amputation of fingers, hands, or feet. Diabetes-related complications are very often the cause of disability. However, proper disease management can help patients maintain a good quality of life [9,10]. Pharmacological therapies are often effective, but they are accompanied by burdensome side effects. The most common are abdominal pain, loss of appetite, nausea, vomiting, digestive disturbances, skin reactions, swelling, and hypoglycemia (blood glucose levels below the physiological range). Additionally, certain drugs cannot be used by patients with the failure of some organs. These limitations significantly reduce conventional therapies in diabetes [13,14]. Therefore, numerous natural, plant-derived compounds continue to be investigated for the development of new drugs [15,16,17,18]. Among multiple agents explored, the anti-diabetic potential of quercetin has been proposed [1,3,19,20]. However, its mechanisms of action are not yet fully understood. In addition, results from animal studies have shown that quercetin alleviates diabetes-related complications, such as kidney failure [21,22,23,24] and cardiovascular problems [25,26,27,28], and exhibits both neuroprotective [29,30,31,32] and hepatoprotective [31,32] effects. Quercetin supplementation has been shown to alleviate diabetic encephalopathy in rats by upregulating the multifunctional regulator nuclear factor erythroid 2-related factor (Nrf2)/heme oxygenase-1 (HO-1) signaling pathway [33]. It also improved cognitive functions in diabetic rats [34]. In addition, quercetin mitigated diabetic nephropathy in rats, likely by inhibiting renal tubular epithelial cell apoptosis via the phosphoinositide 3-kinase (PI3K)/ serine/threonine kinase Akt (also known as protein kinase B) pathway [35,36]. Quercetin also showed a protective effect against pulmonary dysfunction in diabetic rats, possibly through inhibition of the NLRP3 inflammasome signaling pathway [37]. Furthermore, quercetin demonstrated ameliorative potential in limiting diabetic ulcers in the rat model, an effect attributed to its antioxidant and anti-inflammatory properties, as well as to reduced ischemia [38]. Positive effects on reproductive function have also been reported [39], and quercetin has shown promise in mitigating diabetic retinopathy in rats [40]. Based on these findings, quercetin may be a promising candidate for the treatment of diabetes and its complications in humans. However, its potential therapeutic application should be preceded by thorough evaluation of its effects on various pathological processes associated with the disease.

This review summarizes results of in vivo rodent studies, which provide evidence that quercetin effectively mitigates key diabetes-related abnormalities, including hormonal and metabolic disturbances, oxidative and inflammatory stress, and alterations in the expression/action of the relevant enzymes and signaling molecules.

## 2. Effects of Quercetin on Blood Glucose and Insulin Levels and Insulin Resistance

Available data indicate that quercetin administration can reduce blood glucose levels (glycemia). In both healthy people and experimental rodents, glycemia must be maintained within a very narrow physiological range. Under normal conditions, glucose serves as a key energy substrate for various cell types, including neurons, hepatocytes, myocytes, and adipocytes. Insufficient glucose supply can lead to significant disturbances at both the cellular and organ levels. On the other hand, excess glucose delivery in diabetes is associated with cellular damage (glucotoxicity) affecting hepatocytes, myocytes, adipocytes, pancreatic β-cells, neurons, and other cell types. Chronically elevated blood glucose levels contribute to severe diabetes complications, including vascular dysfunction, renal failure, retinopathy, and cardiopathy [9,10,41,42,43,44,45,46,47]. It is well-established that reducing hyperglycemia is essential for the effective management of diabetes and its complications [5,6]. In this context, quercetin was shown to possess blood-glucose-lowering properties, demonstrated in animal models of type 1 and type 2 diabetes.

### 2.1. Type 1 Diabetes

Quercetin supplementation has been shown to reduce hyperglycemia in rats with diabetes induced by streptozotocin (STZ) administration [48,49,50,51,52,53,54]. STZ treatment induces hallmark features of type 1 diabetes in experimental animals by selectively destroying pancreatic β-cells, the only source of circulating insulin [55,56]. This leads to a profound insulin deficiency. Insulin is a crucial hormone regulating many relevant processes, such as the metabolism of carbohydrates, lipids, and proteins, and also intracellular transport. Insulin deficiency is associated with significant metabolic dysregulation. In rats with STZ-induced diabetes, blood insulin levels were significantly reduced (hypoinsulinemia) compared to normal values [48,49,50,51,52,53,54]. Since insulin is the only hormone capable of effectively lowering blood glucose levels, the antihyperglycemic effect of quercetin in these animals is related to an increase in circulating insulin concentrations. The observed rise in blood insulin levels following quercetin treatment indicates a beneficial effect on the endocrine function of pancreatic β-cells. These cells are highly susceptible to various detrimental factors and easily damaged by elevated blood glucose and lipid levels (gluco- and lipotoxicity) [57,58]. The effect of quercetin on insulin-secreting cells in diabetic rodents may be either direct or indirect (see below: Effects of Quercetin on the Pancreas).

### 2.2. Type 2 Diabetes

In addition to its beneficial effects on glycemia in type 1 diabetes, the blood-glucose-lowering properties of quercetin have also been demonstrated in animals with experimentally induced type 2 diabetes. These effects were demonstrated in rats and mice fed a high-energy diet as well as in db/db mice. Rodents maintained on a high-energy diet reveal many symptoms of type 2 diabetes, including obesity, inflammation, hyperglycemia, insulin resistance, and hyperinsulinemia [59,60]. The db/db mouse model carries a spontaneous mutation of the leptin receptor, resulting in obesity and chronic hyperglycemia, and is widely used as a model of type 2 diabetes [61]. Quercetin has been shown to reduce hyperglycemia in rats on a high-fat diet (HFD) and treated with STZ [29,62]. Similar antihyperglycemic effects have been confirmed in db/db mice [63,64,65,66,67] and in mice fed an obesogenic diet [68,69,70]. Moreover, quercetin reduced blood concentrations of hemoglobin A1c (HbA1c) in db/db mice [50,60]. Since HbA1c reflects long-term glycemic control, the observed reduction in HbA1c levels indicates that quercetin exerts sustained blood-glucose-lowering effects [71].

In type 2 diabetes, elevated blood glucose concentrations are primarily associated with insulin resistance. Chronic hyperglycemia and insulin resistance trigger compensatory hypersecretion of insulin, resulting in elevated blood insulin concentrations. However, despite increased blood insulin levels, cells become less responsive to this hormone. Conversely, reducing hyperglycemia is known to improve insulin sensitivity [17,72,73]. In rats with diabetes induced by the administration of STZ and nicotinamide (NA), blood insulin concentrations were indeed raised [74]. In this experimental model, NA is used to attenuate the cytotoxic effects of STZ on pancreatic β-cells, thereby allowing moderate hyperinsulinemia to develop [56]. Quercetin therapy effectively reduced hyperinsulinemia in rats with diabetes induced by STZ/NA [74]. Similar effects were also revealed in mice with hyperinsulinemia induced by a high-fat and high-glucose diet (HFD/HGD) in combination with STZ treatment [70]. The normalization of blood insulin levels in diabetic animals following quercetin administration was associated with improved insulin action.

In type 2 diabetes, prolonged overstimulation of β-cells by excessive glucose supply leads to their exhaustion, progressive failure, and the consequent hypoinsulinemia. In this case, insulin action becomes insufficient [12,75]. In rats with diabetes induced by feeding an HFD and treated with STZ, blood insulin levels were reduced, while quercetin treatment restored concentrations of insulin to normal values [29,62]. Similar effects were observed in other diabetic models characterized by hypoinsulinemia, i.e., HFD/STZ-treated mice [68], HFD/HGD STZ-treated mice [70], and db/db mice [66].

In addition to normalizing blood insulin levels, studies on diabetic rodents with insulin resistance have evidenced that quercetin treatment improves the action of this hormone. Enhanced insulin sensitivity following quercetin administration was observed in several experimental models of insulin resistance: STZ/NA rats [74], db/db mice [64,65], and in mice fed an HFD [69].

Hyperglycemia, abnormal blood insulin levels, and insulin resistance are the main symptoms of type 2 diabetes. Therefore, the normalization of blood glucose and insulin concentrations, along with the improved insulin sensitivity, following quercetin treatment, is critical to the anti-diabetic effects of this compound. The effects of quercetin on blood glucose, insulin levels, and insulin resistance are summarized in Table 1.

## 3. Effects of Quercetin on Blood Lipids

Diabetes is associated with dysregulation of lipid metabolism. These disorders cover various tissues, including the blood [77,78,79]. Studies have revealed that blood levels of non-esterified fatty acids (NEFAs) and triacylglycerides (TGs) are elevated in diabetic rodents compared to healthy controls. This leads to an excessive lipid supply to the insulin-sensitive tissues, such as the liver, skeletal muscle, and adipose tissue, resulting in increased lipid accumulation in these organs. The rise in tissue lipid content largely contributes to impaired insulin action and is one of the main causative factors of insulin resistance [73,80,81]. In rats with STZ-induced diabetes, blood NEFA and TG levels were elevated, while quercetin therapy reduced their concentrations to normal values [23,29,51,52,53,54]. Similar effects of quercetin supplementation on blood TGs were observed in db/db mice [64], and in mice fed an obesogenic diet [69].

Apart from NEFAs and TGs, diabetes-related dysregulation of blood lipids also involves cholesterol. Cholesterol is transported in blood in various lipoprotein fractions. In diabetes, concentrations of total blood cholesterol and low-density lipoprotein (LDL) cholesterol typically increase, whereas high-density lipoprotein (HDL) cholesterol levels are reduced. These disturbances are associated with atherosclerosis, i.e., pathological lesions in the arterial walls [25,78,82]. An elevated total blood cholesterol level has been observed in rats with STZ-induced diabetes [29,50,51,52,54] and in insulin-resistant db/db mice [64]. In each case, quercetin reduced the total cholesterol levels toward values observed in healthy animals. Moreover, in STZ-induced diabetic rats, blood LDL-cholesterol levels were increased; however, this rise was limited in animals subjected to quercetin therapy [32,50,51,52]. Additionally, HDL-cholesterol levels were reduced in rats with STZ-induced diabetes [31,51,62,64], and quercetin reversed these disturbances, restoring HDL-cholesterol concentrations toward normal values [31,51,62,64].

The advantageous action of quercetin on blood lipid profiles points to the ability of this compound to normalize NEFA and TG concentrations and exhibit antiatherogenic properties. These actions contribute to the alleviation of diabetes-related metabolic disturbances. The effects of quercetin on blood lipids are summarized in Table 1.

## 4. Effects of Quercetin on Blood Oxidative and Inflammatory Stress Indices

It is well-established that diabetes is associated with oxidative and inflammatory stress, which also manifests in the blood. Oxidative stress results from excessive formation and/or insufficient removal of reactive oxygen species (ROS), i.e., hydrogen peroxide (H_2_O_2_), hydroxyl radical (OH^·^), and superoxide anion radical (O^·−^). Excessive ROS cause damage to lipids, proteins, and nucleic acids. Diabetes-related oxidative stress contributes to the injury of various kinds of cells, and impairs both insulin secretion and action. The organism possesses several antioxidant defense mechanisms, among which antioxidative enzymes play a key role. These include superoxide dismutase (SOD, EC 1.15.1.1), catalase (CAT, EC 1.11.1.6), and glutathione peroxidase (GPx, EC 1.11.1.9). These enzymes participate in converting ROS into less harmful particles. SOD catalyzes the conversion of superoxide anion radical into oxygen. CAT is involved in the decomposition of hydrogen peroxide into water and oxygen. GPx converts hydrogen peroxide into water [41,45,46,83]. In rats with STZ-induced diabetes, blood SOD, CAT, and GPx activities were reduced, while malondialdehyde (MDA) levels, the end product of lipid peroxidation, were elevated. These alterations indicate a limited capability of the organism to neutralize ROS. Quercetin therapy increased the activity of antioxidant enzymes and concurrently reduced MDA content [32,48,53].

Apart from oxidative stress, diabetes is also accompanied by inflammatory stress, which manifests, among others, in the blood. Excessive production of inflammatory cytokines in diabetes is responsible for numerous harmful changes, such as impaired insulin signaling, insulin resistance, reduced intracellular glucose transport, β-cell failure, and damage to other cell kinds [46]. In rats with STZ-induced diabetes, blood levels of inflammatory markers, tumor necrosis factor-α (TNF-α), and nuclear factor kappa-light-chain-enhancer of activated B cells (NF-κB) were elevated compared to healthy animals, indicating inflammatory stress. However, inflammatory stress was effectively reduced in diabetic rats treated with quercetin [53,76]. In diabetes, oxidative and inflammatory stress interact, exacerbating pathological changes [41,46,84]. Therefore, the antioxidative and anti-inflammatory properties of quercetin demonstrated in experimental animals mitigate diabetes-related disorders. Effects of quercetin on blood oxidative and inflammatory stress indices are summarized in Table 1.

## 5. Effects of Quercetin on the Pancreas

In diabetes, the structure of pancreatic islets and the physiological function of insulin-secreting cells are markedly impaired. These abnormalities occur in both type 1 and type 2 diabetes [9,10,85,86].

### 5.1. Type 1 Diabetes

In rats with diabetes induced by streptozotocin (STZ), pancreatic tissue damage was observed. However, this effect was attenuated by quercetin treatment [32,49,51,52,54,74,76,87]. STZ-induced diabetic rats also showed a reduced number and area of pancreatic islets and β-cells. Quercetin treatment increased both these parameters [53,54,76].

A major factor contributing to β-cell damage in diabetes is oxidative stress. In pancreatic β-cells, the antioxidant defense system is very weak compared to other cell types, making them highly susceptible to ROS-induced injury [55,75]. In rats with diabetes induced by the administration of STZ, pancreatic activities of antioxidant enzymes (CAT, SOD, and GPx) were reduced, while MDA levels were elevated. Quercetin therapy restored antioxidant enzyme activity and decreased MDA levels [49,76]. Pancreatic insulin content was also reduced in STZ-induced diabetic rats; quercetin therapy increased insulin levels in the pancreatic tissue [49], suggesting enhanced insulin secretory capacity of β-cells following treatment.

Pathological alterations developing in diabetes are associated with enhanced pancreatic islet cell autophagy, i.e., lysosome-dependent removal of damaged cellular components [88,89]. Consistent with this, autophagy levels in the pancreatic islets of rats with STZ-induced diabetes were higher than in healthy controls. However, quercetin therapy effectively reduced this process. Suppression of this process appears to play a significant role in the anti-diabetic action of quercetin, as autophagy inhibitors were shown to block its beneficial effects. In that study, rats were treated with 3-methyladenine (3-MA), an autophagy inhibitor that blocks autophagosome formation by inhibiting type III phosphatidylinositol 3-kinases (PI3K). 3-MA is a membrane-permeable compound commonly used to investigate autophagy mechanisms [76].

### 5.2. Type 2 Diabetes

Results of rodent studies have shown that insulin resistance is associated with a reduced number of pancreatic islets and deteriorated islet structure. In hypertensive, insulin-resistant rats, the number of pancreatic islets was markedly diminished compared to non-diabetic animals. Importantly, this detrimental effect was limited due to quercetin therapy [90]. Moreover, in mice on a high-fat diet (HFD), quercetin enhanced pancreatic histology, islet structure, and increased insulin content [68]. Similar effects were observed in mice on a high-fructose/high-glucose diet (HFD/HGD) and treated with STZ [70].

In rodent models of type 2 diabetes, pancreatic cells exhibit symptoms of oxidative stress. In the pancreatic tissue of rats with HFD-induced insulin resistance, the activity of SOD, CAT, and GPx was diminished, while MDA content was elevated. These disturbances were ameliorated by quercetin treatment [52]. Such actions of quercetin on SOD, CAT, and GPx activity and on MDA content were observed in HFD-fed mice [58] and insulin-resistant db/db mice [66].

Another relevant aspect of the anti-diabetic properties of quercetin concerns the regulation of apoptosis, i.e., programmed cell death. Excessive pancreatic cell apoptosis was observed in mice fed an HFH/HGD and treated with STZ, resulting in a reduced number of islet cells. However, quercetin therapy reversed these diabetes-related changes by decreasing the number of apoptotic cells [70]. In addition, in db/db mice, quercetin inhibited apoptosis in the pancreas and simultaneously upregulated the expression of Sirt 3, a factor silencing apoptosis-related genes [66].

Diabetes-related oxidative stress is known to be accompanied by inflammatory stress, and both factors contribute substantially to the progressive β-cell damage [55,91]. In rats fed an HFD, pancreatic content of inflammatory markers (TNF-α and IL-6, IL-1β, and caspase-3) were elevated. These changes were attenuated in diabetic animals subjected to quercetin therapy [62].

Under physiological conditions, proper glucose transport and metabolism in pancreatic β-cells are essential for stimulating insulin secretion. Glucose transport into β-cells is mediated by glucose transporter-2 (GLUT2). Then, glucose is metabolized through oxidative glycolysis. The first step in glucose metabolism is its phosphorylation by glucokinase (EC 2.7.1.2). These processes are impaired in diabetes [92,93]. Indeed, in rats fed an HFD, pancreatic GLUT2 content and glucokinase activity were reduced. However, these pathological changes were mitigated following quercetin supplementation [62].

Pancreatic β-cells failure or damage in diabetes is a major contributor to insulin deficiency. Quercetin therapy has been associated with multiple beneficial effects on insulin-secreting cells, markedly enhancing their secretory function. The effects of quercetin on the pancreas are summarized in Table 2.

## 6. Effects of Quercetin on the Insulin-Sensitive Tissues

### 6.1. The Liver

The liver is one of the insulin-sensitive tissues most adversely affected by diabetes [94,95]. In experimental diabetes, the liver structure was shown to be damaged. Such effects were observed in rats with diabetes induced by the administration of STZ. Quercetin therapy nevertheless improved the liver structure [31,50,87,96]. This is a relevant finding as structural damage in the liver often coincides with functional impairment. The liver dysfunction in diabetes manifests, among others, in excessive glucose output into the blood. In type 1 diabetes, this is driven by reduced blood insulin content; in type 2 diabetes, it results from insulin resistance. In each case, excessive hepatic glucose output largely contributes to elevated blood glucose levels [95,97,98]. In diabetes, liver cells (hepatocytes) reveal metabolic dysregulation. One of the defects concerns glycogen stores. In rats with diabetes induced by STZ and STZ/NA administration, the hepatic glycogen content was depleted [50,74]. This effect was associated with abnormal changes in the activities of enzymes related to glucose metabolism, i.e., increased hepatic glucose-6-phosphatase (EC 3.1.3.9) and glycogen phosphorylase (EC 2.4.1.1). Glucose-6-phosphatase is responsible for the dephosphorylation of glucose-6-phosphate, while glycogen phosphorylase catalyzes glycogen decomposition to glucose. Excessive activity of these enzymes facilitates hepatic glucose release to the blood and contributes to diabetic hyperglycemia [99]. Quercetin supplementation reversed pathological alterations since it downregulated the action of hepatic glucose-6-phosphatase and glycogen phosphorylase [32,74]. Another key enzyme involved in hepatic glycogen metabolism and affected in diabetes is glycogen synthase kinase-3β (GSK3β, EC 2.7.1.37). Its function is, among others, to inhibit glycogen synthase (EC 2.4.1.11) [100]. In rats with STZ-induced diabetes, GSK3β was upregulated, while quercetin supplementation normalized the activity of this enzyme [50]. This beneficial effect also helps preserve hepatic glycogen stores in diabetes.

Moreover, the activity of phosphoenolpyruvate carboxykinase (PEPCK, EC 4.1.1.32), an enzyme involved in glucose synthesis from non-carbohydrate substrates, was shown to be increased in STZ-induced diabetic rats. Upregulation of this enzyme contributes to elevated hepatic glucose output into the blood and the resulting hyperglycemia. However, quercetin therapy restored PEPCK activity to normal levels [32,99].

Apart from pathological disturbances in glucose metabolism, diabetes is also associated with dysregulation of liver lipid metabolism, partly due to the dysfunction of the enzymes involved in lipid homeostasis. In rats with STZ-induced diabetes, hepatic expression and activity of silent information regulator1 (SIRT1) were downregulated, while sterol regulatory element-binding protein-1 (SREBP-1) was upregulated. These alterations were prevented following administration of quercetin [50]. There is an inverse relationship between SIRT1 and SREBP-1 levels. SIRT1, an NAD^+^-dependent deacetylase, is vital in controlling many intracellular processes. In hepatocytes, both SIRT1 and SREBP-1 are involved in the regulation of lipid metabolism. Dysregulation of their activity leads to excessive lipid accumulation and to the resulting hepatic insulin resistance [101,102]. Indeed, liver lipid content was elevated in diabetic animals compared to healthy controls. The concentrations of TGs and NEFAs were increased in rats with STZ-induced diabetes and mice fed an HFD. Quercetin therapy normalized liver lipid accumulation [69], thereby contributing to improved insulin sensitivity in hepatic tissue.

Another relevant effect of quercetin related to improved insulin signaling concerns insulin receptor substrate 1 (IRS-1), one of the key intracellular molecules involved in insulin action. In STZ-induced diabetic rats, hepatic IRS-1 expression was downregulated, and IRS-1 was phosphorylated at the serine 616 residue [54]. Both changes are detrimental to hepatocytes since they are associated with reduced insulin sensitivity [99,103]. However, quercetin therapy restored IRS-1 expression and phosphorylation levels to those observed in healthy animals [54]. In the same experimental model, i.e., in the STZ-induced diabetic rat, hepatic levels of phosphoinositide 3-kinase (PI3K, 2.7.1.137) [54] and phosphorylation of Akt (PKB) [50] were diminished. These enzymes are essential components of the insulin signaling pathway, and their dysregulation contributes to impaired insulin sensitivity [95,104]. Quercetin treatment reversed both pathological changes in PI3K and Akt, thereby improving insulin signaling and enhancing insulin action in the liver [50,54]. However, the cited studies did not specify whether these effects involved transcriptional regulation, post-transcriptional modulation, or direct inhibition of serine phosphorylation. Further investigation is needed to clarify the underlying mechanism.

Oxidative and inflammatory stress in the insulin-sensitive tissues also contributes to diabetes-related insulin resistance and tissue damage [41,46]. In rats with STZ-induced diabetes, the content of malondialdehyde (MDA), a marker of lipid peroxidation, was elevated in hepatic tissue. However, this detrimental effect was alleviated by quercetin treatment [31,74,96]. Diabetes-related oxidative stress was accompanied by the decreased hepatic activity of SOD and CAT, as well as the reduced content of glutathione (GSH). Importantly, quercetin normalized the activity of the antioxidant enzymes and restored the GSH levels [31,74]. Similar changes in MDA content and antioxidant enzyme activity were confirmed in db/db mice [64] and in insulin-resistant hypertensive rats [90].

Along with its beneficial effects on oxidative stress, quercetin has also been shown to alleviate inflammatory stress. In rats with STZ-induced diabetes, hepatic expression of inflammatory markers was elevated. This harmful effect was abolished by quercetin treatment [31,48]. These favorable changes contribute to improved insulin sensitivity and mitigation of diabetes-related liver dysfunction.

Another important indicator of liver dysfunction in diabetes involves enzymatic markers. Blood activities of aspartate aminotransferase (AST, EC 2.6.1.1), alanine aminotransferase (ALT, EC 2.6.1.2), and alkaline phosphatase (ALP, EC 3.1.3.1) were elevated in rats with diabetes induced by STZ alone [32,74] or by STZ combined with an HFD [62]. Under physiological conditions, these enzymes act intracellularly; however, liver cell damage causes their release into the blood. Increased AST, ALT, and ALP activity therefore indicates liver damage [105]. Notably, quercetin supplementation restored the activities of these enzymes to normal levels in diabetic rats [32,62,74].

The effects of quercetin on the liver are summarized in Table 3. They indicate that this compound exerts hepatoprotective effects, thereby alleviating diabetes-related liver dysfunction, including insulin resistance.

### 6.2. The Skeletal Muscle

The skeletal muscle is another insulin-sensitive tissue negatively affected by diabetes [106,107]. One of the key defects includes structural disturbances, which are associated with impaired muscle function. Histological changes have indeed been observed in rats with diabetes induced by the administration of STZ. However, quercetin treatment effectively mitigated these structural abnormalities [108], suggesting a potential improvement in muscle function.

Under physiological conditions, the skeletal muscle is responsible for the majority of glucose uptake from the blood after a meal, contributing to postprandial antihyperglycemic effects. Intramuscular glucose transport is primarily stimulated by insulin. However, insufficient insulin action is associated with diminished muscle glucose uptake and, consequently, hyperglycemia [106,107]. One of the key diabetes-related defects in skeletal muscle is impaired insulin signaling. In sucrose-fed (SF) rats with insulin resistance, phosphorylation of Akt (PKB)—a central molecule in the insulin signaling pathway—was reduced compared to normal animals. This defect contributes to skeletal muscle insulin resistance. However, quercetin therapy restored Akt phosphorylation to the levels observed in control animals [109].

Another relevant factor in skeletal muscle that is disturbed in diabetes is antioxidant defense. In mice with diabetes induced by the administration of alloxan (ALX), skeletal muscle showed reduced activities of antioxidative enzymes, SOD and CAT, and a concomitant decrease in GSH content [110]. In this experimental model, ALX induces destruction of the pancreatic β-cells leading to hallmarks of type 1 diabetes, such as insulin deficiency, marked hyperglycemia, and dysfunction of various tissues, including skeletal muscle [55]. However, quercetin treatment abolished all pathological alterations in antioxidant defense in the skeletal muscle of ALX-induced diabetic mice [110].

Quercetin also positively affected diabetes-related disorders in the activities of other enzymes in skeletal muscle. In mice with diabetes induced by administration of ALX, the activity of hexokinase (EC 2.7.1.1), the enzyme catalyzing the phosphorylation of hexoses, was reduced, which reflects limited intracellular metabolism of hexoses. Nevertheless, quercetin therapy restored hexokinase activity to normal levels [110]. Another enzyme regulated by quercetin in the skeletal muscle is fructose 1,6-bisphosphatase (EC 3.1.3.11), a rate-limiting enzyme that hydrolyzes fructose 1,6-bisphosphate to fructose 6-phosphate and inorganic phosphate. In diabetes, the action of fructose 1,6-bisphosphatase is usually affected, as shown by its elevated activity in the muscle tissue of rats with ALX-induced diabetes. This alteration was, however, normalized following quercetin therapy [110].

Taken together, these findings indicate that quercetin alleviates diabetes-related dysregulation of key skeletal muscle enzymes involved in hexose metabolism.

Additional relevant effects of the anti-diabetic action of quercetin in skeletal muscle concern intracellular glucose transport. Effective glucose uptake into myocytes occurs in the presence of insulin, and is mediated by the plasma membrane transporter-4 (GLUT4). In mice with ALX-induced diabetes, skeletal muscle GLUT4 expression was downregulated. Quercetin therapy restored its expression to the normal level [110].

Collectively, these findings indicate that quercetin markedly alleviates diabetes-associated disturbances in skeletal muscle. This is a particularly important property of quercetin, given the critical role of skeletal muscle in the pathogenesis of diabetes, as well as insulin resistance. The effects of quercetin on skeletal muscle are summarized in Table 4.

### 6.3. The Adipose Tissue

Apart from the liver and skeletal muscle, adipose tissue is one of the insulin-sensitive tissues whose function is significantly impaired in diabetes. Fat tissue has two pivotal functions: it stores energy (in the form of TGs) and secretes adipokines. After a meal, TGs are synthetized in fat cells (adipocytes) in the process of lipogenesis to store the excess energy. In the post-absorptive state, TGs undergo lipolysis, breaking down into glycerol and NEFAs. These products are then released from adipocytes and serve as an energy source for other kinds of cells. Under physiological conditions, a balance between lipogenesis and lipolysis maintains appropriate adipose tissue content. However, excessive fat accumulation is associated, among others, with obesity, metabolic disorders, insulin resistance, and type 2 diabetes. Diabetes-related adipose tissue dysfunction is primarily characterized by enlarged adipocytes, increased production of pro-inflammatory adipokines, and oxidative stress. Importantly, adipose tissue insulin resistance is one of the key hallmarks of type 2 diabetes [111,112,113,114,115]. These pathological alterations can be alleviated by quercetin therapy. In insulin-resistant hypertensive rats, quercetin was shown to increase insulin sensitivity in adipose tissue [90]. In Zucker Diabetic Fatty (ZDF) rats with insulin resistance, quercetin reduced markers of inflammation (macrophage infiltration marker F4/80 and TNF-α) and oxidative stress (nicotinamide adenine dinucleotide phosphate oxidase 2, Nox2, and superoxide dismutase) in adipose tissue. In the same model, quercetin downregulated the expression of genes involved in adipogenesis (GLUT4 and fatty acid binding protein), which was accompanied by reduced adipocyte size [116]. Moreover, in mice with obesity induced by feeding an HFD, quercetin reduced adipose tissue mass [69]. The observed reduction in fat tissue accumulation in both ZDF rats and HFD-fed mice further confirms the ability of quercetin to effectively improve insulin action.

Diabetes-related abnormal adipokine secretion and action are associated with dysregulation of several essential processes, including feeding behavior, energy expenditure, insulin sensitivity, reproductive function, and more. Two adipokines with pivotal regulatory roles that are particularly affected in diabetes are adiponectin and leptin. Adiponectin, an adipocyte-derived hormone, works synergistically with insulin. It promotes intracellular degradation of fatty acids (β-oxidation), thus preventing excessive lipid accumulation within cells. Moreover, adiponectin improves insulin action in insulin-sensitive tissues. However, excessive blood adiponectin levels (hyperadiponectinemia) contribute to its reduced action and are associated with insulin resistance [117,118,119]. In insulin-resistant db/db mice, elevated blood adiponectin levels along with adiponectin resistance were observed. Nevertheless, quercetin supplementation normalized adiponectin concentrations in these animals [64].

Another relevant adipokine involved in insulin action is leptin. Circulating leptin originates primarily from adipocytes. This hormone exerts anorexigenic effects (reduces appetite), increases energy expenditure, and enhances insulin sensitivity. However, in the case of prolonged hyperleptinemia (elevated blood leptin levels), target cells develop leptin resistance, which contributes to impaired insulin action [117,118,119]. Hyperleptinemia was observed in obese mice fed an HFD, whereas quercetin therapy reduced blood leptin levels to normal values [59]. These quercetin-induced effects on blood adiponectin and leptin levels suggest improved adipokine function in diabetic animals.

Quercetin-induced changes in adipose tissue are associated with the alleviation of diabetes-related disturbances, such as excessive fat accumulation, impaired insulin action, and elevated blood adipokine levels. The effects of quercetin on adipose tissue are summarized in Table 5.

## 7. Additional Remarks

Insulin resistance is the central hallmark of type 2 diabetes. This multifactorial defect affects insulin-sensitive tissues, i.e., the skeletal muscle, the liver, and the adipose tissue. In each of these tissues, impaired insulin action arises from a combination of metabolic disturbances, oxidative stress, and inflammatory stress. Moreover, insulin resistance is also strongly associated with pathological changes in blood parameters, which further diminish the action of the pancreatic hormone. All these alterations influence each other and are interconnected; beneficial or pathological changes in one parameter induce changes in the others. Given the complexity and interconnected nature of these abnormalities, therapeutic strategies capable of targeting multiple pathological processes simultaneously are particularly valuable. Numerous studies have documented that quercetin simultaneously mitigates all pathological lesions (not just a single parameter) in blood and insulin-sensitive tissues, thereby improving insulin sensitivity.

Despite its potential, quercetin has limited water solubility, which may reduce its use in anti-diabetic therapies. Furthermore, the native form of the compound exhibits relatively low bioavailability after ingestion. However, converting native quercetin into nano-formulation or liposomal formulation markedly increases its absorption, bioavailability, and therapeutic potential. The critical feature of quercetin is its lipophilicity, which allows it to readily penetrate cellular membranes. This characteristic suggests that quercetin may act by both extracellular and intracellular mechanisms [120,121].

Due to poor absorption from the digestive tract, some natural, bioactive compounds must be administered via injections to be efficacious. In contrast, quercetin was active in anti-diabetic therapies when given intraperitoneally, orally, or in a diet. The effectiveness after oral delivery is highly relevant to the potential use of quercetin as a drug.

Another relevant issue concerns the doses used in research. Quercetin has been proven safe in humans. Even at doses of up to 5000 mg per day, no adverse effects have been reported [1,3]. Rodent studies demonstrated that the lowest effective anti-diabetic dose of quercetin was 10 mg/kg b.w. [50]. Pharmacokinetics research in humans has also revealed that pure quercetin is absorbed from the digestive tract and reaches the blood. Notably, after a single intragastric dose, quercetin remains detectable in the blood for many hours (at least 12 h after administering 250–1000 mg). This prolonged presence, compared to many other bioactive compounds, is primarily due to enterohepatic recycling. These findings also suggest that quercetin may achieve therapeutic effects in humans when administered once or twice daily. The bioavailability of quercetin in humans was significantly higher when delivered in lipomicellar formulations compared to the native compound [122,123]. Similarly, in rodent studies on the anti-diabetic effects of quercetin, nano-formulated or liposomal preparations were used to enhance therapeutic efficacy. In humans, the main metabolites detected in the blood were methylated, sulfate, and glutathione (GSH) conjugates of quercetin. The same human trial demonstrated that quercetin did not induce any side effects after oral administration of 1000 mg per person [122,123]. Given that the lowest effective anti-diabetic dose in rats was 10 mg/kg b.w., the corresponding dose for an average human (70 kg) would be approximately 700 mg. This dose is considered safe.

Another important finding is that quercetin did not induce side effects in diabetic rodents, even after long-term supplementation. The longest documented anti-diabetic therapy in rats (50 mg/kg) lasted 12 weeks [50]. Regardless of the experimental model used, quercetin treatment was associated with normalization of pathologically altered parameters. One example may be its effect on blood insulin concentrations. Whether hyperinsulinemia or hypoinsulinemia was present, quercetin restored insulin levels to normal (normoinsulinemia).

Moreover, other studies have evidenced that quercetin effectively alleviates diabetes-related complications (see Introduction). These complications result from hormonal and metabolic disturbances and oxidative and inflammatory stress. There is a bidirectional relationship between such complications and the aforementioned disturbances. Quercetin was shown to normalize these underlying factors, which may contribute to the alleviation of diabetes complications.

It should also be noted that quercetin has many derivatives, such as quercitrin, pentamethylquercetin, dihydroquercetin, isoquercetin, rutin, hyperoside, and avicularin, which have also demonstrated anti-diabetic properties [124,125,126,127,128,129].

Given that type 2 diabetes is usually associated with overweight or obesity, the effects of quercetin on body weight in experimental animals were explored. Some studies evidenced that in obese models of type 2 diabetes, quercetin therapy eliminated pathological alterations without producing detectable changes in body weight gain [63,65,90,116]. This indicates that reduced adiposity is not a prerequisite for quercetin to exert its anti-diabetic effects. This is an important finding since in people, type 2 diabetes is not always associated with obesity.

Taken together, the preclinical data on safety, pharmacokinetics, and efficacy of quercetin highlight its potential for translation into clinical settings. These findings warrant further evaluation in humans, which is discussed in the concluding section below.

## 8. Conclusions

Results of rodent studies have shown that quercetin exerts potent anti-diabetic effects. Its therapeutic efficacy was proven in models of both type 1 and type 2 diabetes. Given the complex and multifactorial nature of diabetes, encompassing interconnected disturbances, effective treatment should target all aspects of the disease. Notably, quercetin exhibited pleiotropic action by attenuating all pathological lesions. Quercetin effectively mitigated diabetes-related hyperglycemia, normalized blood insulin and lipid levels, increased insulin action, improved pancreas structure and endocrine function, reduced oxidative and inflammatory stress, and ameliorated dysfunction in the liver, skeletal muscle, and adipose tissue. These effects were mediated through the modulation of various regulatory enzymes and signaling molecules.

The anti-diabetic properties of quercetin have been consistently proven in various animal models, i.e., diabetes induced by streptozotocin, alloxan, high-energy diet, and determined genetically. In light of this comprehensive preclinical evidence, quercetin emerges as a promising candidate for further translational research. Its consistent efficacy across diverse rodent models, coupled with a favorable safety profile and oral bioavailability, provides a strong rationale for clinical evaluation in humans.

## Figures and Tables

**Figure 1 ijms-26-07391-f001:**
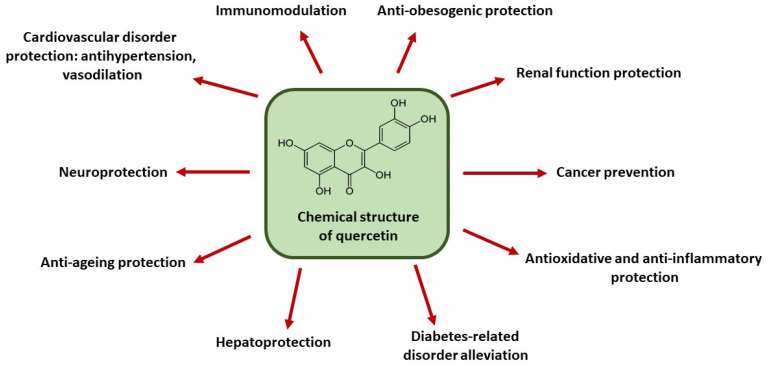
Health-promoting effects of quercetin.

**Table 1 ijms-26-07391-t001:** Quercetin effects on rodent diabetes-related blood parameters, blood lipids, and blood oxidative and inflammatory stress indices.

Parameter	Effect	Animal Model of Diabetes	References
Hyperglycemia	↓	rats with STZ-induced diabetes;	[48,49,50,51,52,53,54]
		HFD/STZ rats;	[29,62]
		db/db mice;	[63,64,65,66,67]
		mice fed an obesogenic diet	[68,69,70]
Hemoglobin A1c	↓	db/db mice	[50,63]
Hyperinsulinemia	↓	rats with STZ/NA-induced diabetes;	[74]
		HFD/STZ, HGD/STZ mice	[70]
Hypoinsulinemia	↑	rats with STZ-induced diabetes;	[48,49,50,51,52,53,54]
		HFD/STZ rats;	[29,60]
		HFD/STZ mice;	[68]
		HFD/STZ, HGD/STZ mice;	[70]
		db/db mice	[66]
Insulin sensitivity	↑	rats with STZ/NA-induced diabetes;	[74]
		HFD mice;	[69]
		db/db mice	[64,65]
Non-esterified fatty acids (NEFA)	↓	rats with STZ-induced diabetes	[23,29,51,52,53,54]
Triglycerides (TG)	↓	rats with STZ-induced diabetes;	[23,29,51,52,53,54]
		db/db mice;	[64]
		mice fed an obesogenic diet	[69]
Total cholesterol	↓	rats with STZ-induced diabetes;	[29,50,51,52,54]
		db/db mice	[64]
LDL-cholesterol	↓	rats with STZ-induced diabetes	[32,50,51,52]
HDL-cholesterol	↑	rats with STZ-induced diabetes;	[32,51,62]
		db/db mice	[64]
Catalase (CAT), superoxide dismutase (SOD), and glutathione peroxidase (GPx) activity	↑	rats with STZ-induced diabetes	[32,48,53]
Malondialdehyde (MDA)	↓	rats with STZ-induced diabetes	[32,48,53]
TNF-α, NF-κB	↓	rats with STZ-induced diabetes	[53,76]

↑—increase, ↓—decrease.

**Table 2 ijms-26-07391-t002:** Quercetin effects on pancreas in diabetic rodents.

Parameter	Effect	Animal Model of Diabetes	References
Tissue structure	↑	rats with STZ-induced diabetes	[32,49,51,52,54,74,76,87]
Area of pancreatic islets and number of β-cells	↑	rats with STZ-induced diabetes	[53,54,76]
Number of pancreatic islets	↑	rats with STZ-induced diabetes;	[53,54,76]
		hypertensive, insulin-resistant rats	[90]
Pancreas histology	↑	HFD mice;	[68]
		HFD/STZ, HGD/STZ mice	[70]
Islet structure	↑	HFD mice;	[68]
		HFD/STZ, HGD/STZ mice	[70]
Catalase (CAT), superoxide dismutase (SOD), and glutathione peroxidase (GPx) activity	↑	rats with STZ-induced diabetes;	[49,76]
		HFD rats;	[62]
		HFD, mice;	[68]
		db/db mice	[66]
Malondialdehyde (MDA)	↓	rats with STZ-induced diabetes;	[49,76]
		HFD rats;	[62]
		HFD mice;	[68]
		db/db mice	[66]
Pancreatic insulin content	↑	rats with STZ-induced diabetes;	[49]
		HFD/STZ, HGD/STZ mice;	[70]
		HFD mice	[68]
Autophagy	↓	rats with STZ-induced diabetes	[76]
Apoptosis	↓	HFD/STZ, HGD/STZ mice;	[70]
		db/db mice	[66]
TNF-α; IL-6; IL-1β; caspase-3	↓	HFD rats	[62]
GLUT2 content; glucokinase activity	↓	HFD rats	[62]

↑—increase, ↓—decrease.

**Table 3 ijms-26-07391-t003:** Quercetin effects on the liver in diabetic rodents.

Parameter	Effect	Animal Model of Diabetes	References
Tissue structure	↑	rats with STZ-induced diabetes	[31,50,87,96]
Glucose-6-phosphatase activity; glycogen phosphorylase activity	↓	rats with STZ-induced diabetes	[32,74]
Glycogen synthase kinase-3β (GSK3β) activity	↓	rats with STZ-induced diabetes	[50]
Phosphoenolpyruvate carboxykinase (PEPCK) activity	↓	rats with STZ-induced diabetes	[32,99]
Silent information regulator1 (SIRT1) expression and activity	↑	rats with STZ-induced diabetes	[50]
Sterol regulatory element-binding protein-1 (SREBP-1) expression	↓	rats with STZ-induced diabetes	[50]
Non-esterified fatty acids’ (NEFA) content	↓	HFD mice	[69]
Insulin receptor substrate 1 (IRS-1) expression and phosphorylation	↑	rats with STZ-induced diabetes	[54]
Phosphoinositide 3-kinase (PI3K) activity;	↑	rats with STZ-induced diabetes	[54]
protein kinase B (Akt/PKB) activity			[50]
Inflammatory markers	↓	rats with STZ-induced diabetes	[48,49]
Catalase (CAT) and superoxide dismutase (SOD) activity	↑	rats with STZ-induced diabetes;	[31,74]
		db/db mice;	[64]
		insulin-resistant hypertensive rats	[90]
Reduced glutathione (GSH) content	↑	rats with STZ-induced diabetes	[31,74]
Malondialdehyde (MDA)	↓	rats with STZ-induced diabetes;	[31,74,96]
		db/db mice;	[64]
		insulin-resistant hypertensive rats	[90]
Aspartate aminotransferase (AST), alanine aminotransferase (ALT), and alkaline phosphatase (ALP) activity	↓	rats with STZ-induced diabetes;	[32,74]
		HFD/STZ rats	[62]

↑—increase, ↓—decrease.

**Table 4 ijms-26-07391-t004:** Quercetin effects on skeletal muscle in diabetic rodents.

Parameter	Effect	Animal Model of Diabetes	References
Structural disorders	↑	rats with STZ-induced diabetes	[108]
Hexokinase activity	↑	mice with ALX-induced diabetes	[110]
Fructose 1,6-bisphosphatase activity	↓	mice with ALX-induced diabetes	[110]
Protein kinase B (Akt/PKB) activity	↑	SF—insulin-resistant rats	[109]
Catalase (CAT) and superoxide dismutase (SOD) activity	↑	mice with ALX-induced diabetes	[110]
Glutathione (GSH) content	↑	mice with ALX-induced diabetes	[110]
GLUT4 expression	↑	mice with ALX-induced diabetes	[110]

↑—increase, ↓—decrease.

**Table 5 ijms-26-07391-t005:** Quercetin effects on adipose tissue and adipokines dysregulation in diabetic rodents.

Parameter	Effect	Animal Model of Diabetes	References
Tissue mass	↓	HFD mice;	[69]
		ZDF rats	[116]
Adipocyte size	↓	ZDF rats	[116]
Insulin sensitivity	↑	insulin-resistant hypertensive rats	[90]
Macrophage infiltration marker F4/80 expression; tumor necrosis factor α	↓	ZDF rats	[116]
Nicotinamide adenine dinucleotide phosphate oxidase 2 (Nox2) expression; superoxide dismutase (SOD) expression	↓	ZDF rats	[116]
Expression of GLUT4 and fatty acid binding protein	↓	ZDF rats	[116]
Adiponectin resistance	↓	db/db mice	[64]
Hyperleptinemia	↓	HFD mice	[69]

↑—increase, ↓—decrease.

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
