# Peer review of "Quercetin as an Anti-Diabetic Agent in Rodents—Is It Worth Testing in Humans?"

_ijms, 2025, doi:10.3390/ijms26157391_

Round 1

Reviewer 1 Report

Comments and Suggestions for Authors

It is a fantastic review and well written. It focus on the potential application of  Quercetin as an anti-diabetic agent. Almost all aspects were introduced and well illustated. It is very friendly to readers and could be accepted after the following points were addressed.

  1. The literatures should be well formated. By the way, is the DOI number necessary?What is the matter in line 642?
  2. The chemical derivatives of  Quercetin might also play a key role in anti-diabetic agents. The authors should include the corresponding apspect as well.

Author Response

Response to Reviewer 1:

Szkudelski et al.

ijms-3783268

We would like to thank the Reviewer for the valuable time and constructive feedback, which significantly helped to improve the quality and clarity of our manuscript. Below, we provide a detailed point-by-point response to each Comment. All changes have been incorporated into the revised version of the manuscript and highlighted using the "Track Changes" function.

General comment: It is a fantastic review and well written. It focus on the potential application of  Quercetin as an anti-diabetic agent. Almost all aspects were introduced and well illustated. It is very friendly to readers and could be accepted after the following points were addressed.

Comment 1:The literatures should be well formated. By the way, is the DOI number necessary?What is the matter in line 642?

Comment 2: The chemical derivatives of  Quercetin might also play a key role in anti-diabetic agents. The authors should include the corresponding aspect as well.

Response 2: We agree with the Reviewer. The following information has been added in the Additional Remarks section, page 17, line 559-561:

“It should also be noted that quercetin has many derivatives, such as quercitrin, pentamethylquercetin, dihydroquercetin, isoquercetin, rutin, hiperoside, and aviku-larin, which have also demonstrated anti-diabetic properties [124-129].”

The following references have been added:

Yong, P.H.; Qing, T.Y.; Azzani, M.; Anbazhagan, D.; Ng, Z.X. Role of medicinal plants in ameliorating the lipid and glucose levels in diabetes: A systematic literature review.  Endocr. Regul. 2025, 59, 57-77.

Liu, Y.; Sun.; Z, Dong, R.; Liu, P.; Zhang, X.; Li, Y.; Lai, X.; Cheong, H.F.; Wu, Y.; Wang, Y.; Zhou, H.; Gui, D.; Xu, Y. Rutin ameliorated lipid metabolism dysfunction of diabetic NAFLD via AMPK/SREBP1 pathway. Phytomedicine, 2024, 126, 155437.

Lazuardi, M.; Anjani, Q.K.; Budiatin, A.S.; Restiadi, T.I. Efficacy of quercetin-like compounds from the mistletoe plant of Dendrophthoe pentandra L. Miq, as oral random blood sugar lowering treatment in diabetic rats. Vet. Q. 2024, 44, 1-14.

Trakooncharoenvit, A.; Hara, H.; Hira, T. Combination of alpha-glycosyl-isoquercitrin and soybean fiber promotes quercetin bioavailability and glucagon-like peptide-1 secretion and improves glucose homeostasis in rats fed a high-fat high-sucrose diet. J. Agric. Food Chem. 2021, 69, 5907-5916.

Rey, D.; Fernandes, T.A.; Sulis, P.M.; Gonçalves, R.; Sepúlveda, R.M.; Silva Frederico, M.J.; Aragon, M.; Ospina.; L.F.; Costa, G.M.; Silva, F.R.M.B. Cellular target of isoquercetin from Passiflora ligularis Juss for glucose uptake in rat soleus muscle. Chem. Biol. Interact. 2020, 330, 109198.

Wang, Y.; Xin, X.; Jin, Z.; Hu, Y.; Li, X.; Wu, J.; Jin, M. Anti-diabetic effects of pentamethylquercetin in neonatally streptozotocin-induced diabetic rats. Eur. J. Pharmacol. 2011, 668, 347-353.

Reviewer 2 Report

Comments and Suggestions for Authors

This review article comprehensively explores quercetin’s anti-diabetic effects in rodent models, detailing its pleiotropic actions on metabolic, oxidative, and inflammatory pathways. The authors present a well-structured synthesis of evidence showing that quercetin mitigates hyperglycemia, modulates insulin levels, and improves function in key insulin-sensitive tissues like the pancreas, liver, skeletal muscle, and adipose tissue. Notably, it acts through modulation of enzymes such as SOD, CAT, GPx, GLUT2, GSK3β, and PEPCK, and signaling proteins like IRS-1 and Akt. The manuscript underscores quercetin’s safety, bioavailability enhancements through nano/liposomal formulations, and encourages human trials. However, major revisions are necessary, particularly regarding recent literature inclusion, figure clarity, mechanistic depth, and balance between summary and critique.

Comments for authors

Comment 1. The authors highlight quercetin’s multiple benefits, but a more recent synthesis of literature from 2023–2025 on its diabetic applications is lacking. A deeper integration of current findings would enrich this section.

Comment 2. The rationale for translating rodent data into human trials needs strengthening. The authors should discuss translational challenges, including dosage differences and metabolic pathways.

Comment 3. A more balanced introduction would benefit from integrating recent pharmacokinetic or therapeutic findings from the last two years to reflect current research momentum.

Comment 4.  The introduction lacks the recent literature on biomedical applications of plant-derived compounds. I recommend the inclusion of a very recent study in the introduction section:

J.N. Rana, S. Mumtaz, Prunin: An Emerging Anticancer Flavonoid, Int. J. Mol. Sci. 26 (2025). https://doi.org/10.3390/ijms26062678.

Comment 5. The authors list improvements in insulin sensitivity, but mechanisms are vaguely described. Clarify how quercetin affects insulin receptor phosphorylation or GLUT4 translocation mechanistically.

Comment 6. The claim that quercetin “increased β-cell number” needs caution. Is this regeneration, neogenesis, or protection from apoptosis? Clarify or adjust the language.

Comment 7. The statement that “autophagy inhibitors suppressed quercetin action” is important. Was this tested directly in vivo? Cite original data and expand the implication.

Comment 8. Hepatic IRS-1 and Akt pathway restoration is crucial, but the mechanistic discussion is very poor. Was this post-transcriptional modulation or direct inhibition of serine phosphorylation?

Comment 9. Throughout the manuscript, the English usage is overly repetitive (“quercetin therapy,” “diabetes-related disturbances”) and sometimes redundant. A careful language edit is recommended to improve clarity and flow.

Author Response

Response to Reviewer 2:

Szkudelski et al.

ijms-3783268

We would like to thank the Reviewer for the valuable time and constructive feedback, which significantly helped to improve the quality and clarity of our manuscript. Below, we provide a detailed point-by-point response to each Comment. All changes have been incorporated into the revised version of the manuscript and highlighted using the "Track Changes" function.

General comment: This review article comprehensively explores quercetin’s anti-diabetic effects in rodent models, detailing its pleiotropic actions on metabolic, oxidative, and inflammatory pathways. The authors present a well-structured synthesis of evidence showing that quercetin mitigates hyperglycemia, modulates insulin levels, and improves function in key insulin-sensitive tissues like the pancreas, liver, skeletal muscle, and adipose tissue. Notably, it acts through modulation of enzymes such as SOD, CAT, GPx, GLUT2, GSK3β, and PEPCK, and signaling proteins like IRS-1 and Akt. The manuscript underscores quercetin’s safety, bioavailability enhancements through nano/liposomal formulations, and encourages human trials. However, major revisions are necessary, particularly regarding recent literature inclusion, figure clarity, mechanistic depth, and balance between summary and critique.

 Comments for authors

Comment 1. The authors highlight quercetin’s multiple benefits, but a more recent synthesis of literature from 2023–2025 on its diabetic applications is lacking. A deeper integration of current findings would enrich this section.

Response 1: We appreciate this suggestion. We have updated the Introduction to include recent findings from 2023 to 2025. These new studies cover various diabetes-related complications mitigated by quercetin, including encephalopathy, nephropathy, retinopathy, testicular dysfunction, diabetic ulcers, and cognitive impairment. Relevant recent references have been added.

The following passage was included in Introduction, page 3, line 89 - 99:

“Quercetin supplementation has been shown to alleviate diabetic encephalopathy in rats by upregulating the Nrf2/HO-1 signaling pathway [33]. It also improved cognitive functions in diabetic rats [34]. In addition, quercetin mitigated diabetic nephropathy in rats, likely by inhibiting renal tubular epithelial cell apoptosis via the PI3K/AKT pathway [35,36].  Quercetin also showed a protective effect against pulmonary dysfunction in diabetic rats, possibly through inhibition of the NLRP3 signaling pathway. [37]. Furthermore, quercetin demonstrated ameliorative potential in limiting diabetic ulcers in the rat model, an effect attributed to its antioxidant and anti-inflammatory properties, as well as to reduced ischemia [38]. Positive effects on reproductive function have also been reported [39], and quercetin has shown promise in mitigating diabetic retinopathy in rats [40].”

The following references have been added:

Zhang, W.; Yi, C.; Song, Z.; Yu, B.; Jiang, X.; Guo, L.; Huang, S.; Xia, T.; Huang, F.; Yan, Y.; Li, H.; Dai, Y. Reshaping the gut microbiota: Tangliping decoction and its core blood-absorbed component quercetin improve diabetic cognitive impairment. Phytomedicine 2025, 140, 156560.

Cappellani, F.; Foti, R.; Malaguarnera, G.; D'Esposito, F.; Musumeci, C.; Rapisarda, L.; Tognetto, D.; Gagliano, C.; Zeppieri, M. Nutrients and natural substances for hypoglycemic effects and management in diabetic retinopathy. Nutrients, 2025, 30, 1207.

Mansour, M.F.; Behairy, A.; Mostafa, M.; Khamis, T.; Alsemeh, A.E.; Ahmed, N.M.Q.; El-Emam, M.M.A. Quercetin-loaded PEGylated liposomes alleviate testicular dysfunction in alloxan-induced diabetic rats: The role of Kisspeptin/Neurokinin B/Dynorphin pathway. Toxicol. Appl. Pharmacol. 2025, 499, 117337.

Larijani, G.; Lotfi, A.; Barati, S.; Janzadeh, A.; Abediankenari, S.; Faghihi, F.; Amini, N. The effect of chitosan/alginate hydrogel loaded quercetin on wound healing in diabetic rat model. J. Mol. Histol. 2025, 56, 225.

Cheng, M.; Yuan, C.; Ju, Y.; Liu, Y.; Shi, B.; Yang, Y.; Jin, S.; He, X.; Zhang, L.; Min, D. Quercetin attenuates oxidative stress and apoptosis in brain tissue of APP/PS1 double transgenic AD mice by regulating keap1/Nrf2/HO-1 pathway to improve cognitive impairment. Behav. Neurol. 2024, 28, 5698119.

Liu, F.; Feng, Q.; Yang, M.; Yang, Y.; Nie, J.; Wang, S. Quercetin prevented diabetic nephropathy by inhibiting renal tubular epithelial cell apoptosis via the PI3K/AKT pathway. Phytother. Res. 2024, 38, 3594-3606.

Chen, P.; Rao, X.; He, P.; Liu, J.; Chu, Y.; Dong, Y.; Ding, M. The role of quercetin in the treatment of kidney diseases: A comprehensive review. Biomed. Pharmacother. 2025, 190, 118358.

El-Shaer, N.O.; Hegazy, A.M.; Muhammad, M.H. Protective effect of quercetin on pulmonary dysfunction in streptozotocin-induced diabetic rats via inhibition of NLRP3 signaling pathway. Environ. Sci. Pollut. Res. Int. 2023, 30, 42390-42398.

Comment 2. The rationale for translating rodent data into human trials needs strengthening. The authors should discuss translational challenges, including dosage differences and metabolic pathways.

Comment 3. A more balanced introduction would benefit from integrating recent pharmacokinetic or therapeutic findings from the last two years to reflect current research momentum.

Response 2/3. We appreciate these insightful comments. Our original manuscript already included a section (Additional Remarks) discussing quercetin’s dosage and bioavailability. However, in response to the Reviewer’s suggestions, we have significantly expanded and updated this section to better highlight the translational relevance of rodent data to potential human applications.

Specifically, we now:

  • clarify the conversion of effective rodent doses into human-equivalent doses, emphasizing that such doses fall well within the safety margins demonstrated in clinical trials;
  • present data from recent human pharmacokinetic studies showing that quercetin is absorbed after oral administration, remains in the bloodstream for many hours due to enterohepatic recycling, and is metabolized into safe conjugated forms;
  • compare native vs. nano- or liposomal formulations, emphasizing the improved bioavailability and therapeutic potential of the latter;
  • reinforce the notion that quercetin does not require invasive delivery methods, as it is effective via oral and dietary routes, which is essential for clinical applicability.

To support these points, new literature from 2023–2025 has been cited (e.g., Solnier et al. 2023; Walle et al. 2001), and the following paragraph has been added to the Additional Remarks section (page 17, lines 530-545):

“Pharmacokinetics research in humans has also revealed that pure quercetin is absorbed from the digestive tract and reaches the blood. Notably, after a single intragastric dose, quercetin remains detectable in the blood for many hours (at least 12 h after administering 250 – 1000 mg). This prolonged presence, compared to many other bioactive compounds, is primarily due to enterohepatic recycling. These findings also suggest that quercetin may achieve therapeutic effects in humans when administered once or twice daily. The bioavailability of quercetin in humans was significantly higher when delivered in lipomicellar formulations compared to the native compound [122,123]. Similarly, in rodent studies on the anti-diabetic effects of quercetin, nano-formulated or liposomal preparations were used to enhance therapeutic efficacy. In humans, the main metabolites detected in the blood were methylated, sulfate, and glutathione (GSH) conjugates of quercetin. The same human trial demonstarted that quercetin did not induce any side effects after oral administration of 1000 mg per person [122,123]. Given that the lowest effective anti-diabetic dose in rats was 10 mg/kg b.w., the corresponding dose for an average human (70 kg), would be approximately 700 mg. This dose is considered safe.”

We have also added a summarizing sentence at the end of the Additional Remarks section (page 17, line 569-571) to explicitly highlight the translational relevance of preclinical data and to ensure a logical transition to the Conclusions section. The following sentence was introduced:

“Taken together, the preclinical data on safety, pharmacokinetics, and efficacy of quercetin highlight its potential for translation into clinical settings. These findings warrant further evaluation in humans, which is discussed in the concluding section below.”   

New references:

Solnier, J.; Zhang, Y.; Roh, K.; Kuo, Y.C.; Du, M.; Wood, S.; Hardy, M.; Gahler, R.J.; Chang, C. A Pharmacokinetic study of different quercetin formulations in healthy participants: A diet-controlled, crossover, single- and multiple-dose pilot study. Evid. Based Complement. Alternat. Med. 2023, 2023, 9727539.

Walle, T.; Walle, U.K.; Halushka, P.V. Carbon dioxide is the major metabolite of quercetin in humans. J. Nutr. 2001, 131, 2648-2652.

We have also decided to transfer the phrase:

“Another relevant issue concerns the doses used in research. Quercetin has been proven safe in humans. Even at doses of up to 5 g per day, no adverse effects have been reported [1,3]. Rodent studies demonstrated that the lowest effective anti-diabetic dose of quercetin was 10 mg/kg b.w. [50].”

to lines 527-530 on 17th page.

Comment 4.  The introduction lacks the recent literature on biomedical applications of plant-derived compounds. I recommend the inclusion of a very recent study in the introduction section:

J.N. Rana, S. Mumtaz, Prunin: An Emerging Anticancer Flavonoid, Int. J. Mol. Sci. 26 (2025). https://doi.org/10.3390/ijms26062678.

Response 4.  According to Reviewer suggestions, the Introduction section has been enriched by new data concerning the biomedical applications of biologically-active compounds (from page 1, line 32 to page 2, line 44):

“Quercetin is one of many biologically active compounds that have attracted significant interest due to their potential therapeutic properties. Biomedical applications of biologically active compounds are of growing interest to scientists and physicians, as well as to patients and health-conscious individuals, as they offer promising possibilities for preventing and treating a wide range of diseases, typically without inducing side effects. Biologically-active agents are known to exhibit properties including antioxidant, anti-inflammatory, antimicrobial, anti-aging, and anti-hypoxia. They may help prevent or treat conditions such as cancer, diabetes, asthma, neurodegenerative disorders, cardiovascular diseases. Some compounds may also regulate the cell cycle, cell proliferation, and apoptosis. Molecular, intracellular targets through which these compounds exert therapeutic effects include: the suppression of pro-inflammatory cytokines, downregulation of transcription factors, and inhibition of some kinases and inducible enzymes [4-7].”

The recent articles have been added to the list of references:

Rana, J.N.; Mumtaz, S. Prunin: an emerging anticancer flavonoid. Int. J. Mol. Sci. 2025, 16, 2678.

Zahra, M.; Abrahamse, H.; George, B.P. Flavonoids: antioxidant powerhouses and their role in nanomedicine. Antioxidants (Basel), 2024, 29, 922.

Gasmi, A.; Asghar, F.; Zafar, S.; Oliinyk, P.; Khavrona, O.; Lysiuk, R.; Peana, M.; Piscopo, S.; Antonyak, H.; Pen, J.J.; Lozynska, I.; Noor, S.; Lenchyk, L.; Muhammad, A.; Vladimirova, I; Dub, N.; Antoniv, O.; Tsal, O.; Upyr, T.; Bjørklund, G. Berberine: pharmacological features in health, disease and aging. Curr. Med. Chem. 2024, 31, 1214-1234.

Tiwari, P.; Mishra, R.; Mazumder, A.; Mazumder, R.; Singh, A. An insight into diverse activities and targets of flavonoids. Curr. Drug Targets. 2023, 24, 89-102.

Comment 5. The authors list improvements in insulin sensitivity, but mechanisms are vaguely described. Clarify how quercetin affects insulin receptor phosphorylation or GLUT4 translocation mechanistically.

Response 5. Thank you for this comment. We agree that a better explanation of the mechanisms by which quercetin affects insulin sensitivity would strengthen the manuscript. Unfortunately, the original studies included in our manuscript did not provide further mechanistic explanations, such as whether the observed effects resulted from post-transcriptional modulation or inhibition of serine phosphorylation.

It is well established that in individuals with type 2 diabetes, insulin resistance is a multifactorial condition involving numerous pathological alterations, particularly in insulin-sensitive tissues and pancreatic β-cells. The question of whether insulin resistance precedes metabolic disturbances, oxidative stress, and inflammation, or results from them, remains open.

The mechanism by which quercetin improves insulin sensitivity is similarly complex and pleiotropic. The reviewed literature (and our manuscript) presents evidence that quercetin alleviates insulin resistance by acting on multiple pathological features simultaneously (including metabolic dysfunctions, oxidative stress, and inflammation) rather than by targeting a single, isolated molecular pathway.

This broader context and the pleiotropic mode of action of quercetin are reflected throughout our manuscript in various sections. Nevertheless, we acknowledge the need for future studies to explore in greater detail the precise intracellular mechanisms by which quercetin influences insulin signaling pathways, including insulin receptor phosphorylation and GLUT4 translocation.

Moreover, the following sentences have been added in the Additional Remarks section, page 16, lines: 504-515:

“Insulin resistance is the central hallmark of type 2 diabetes. This multifactorial defect affects insulin-sensitive tissues, i.e., the skeletal muscle, the liver, and the adipose tissue. In each of these tissues, impaired insulin action arises from a combination of metabolic disturbances, oxidative stress, and inflammatory stress. Moreover, insulin resistance is also strongly associated with pathological changes in blood parameters, which further diminish the action of the pancreatic hormone. All these alterations influence each other and are interconnected; beneficial or pathological changes in one parameter induce changes in the others. Given the complexity and interconnected nature of these abnormalities, therapeutic strategies capable of targeting multiple pathological processes simultaneously are particularly valuable. Numerous studies have documented that quercetin simultaneously mitigates all pathological lesions (not just a single parameter) in blood and insulin-sensitive tissues, thereby improving insulin sensitivity.”    

Comment 6. The claim that quercetin “increased β-cell number” needs caution. Is this regeneration, neogenesis, or protection from apoptosis? Clarify or adjust the language.

Response 6. We appreciate this point. In each of the references cited, the studies reported that quercetin increased the number and area of pancreatic islets and β-cells. However, they did not explore whether this effect was due to regeneration, neogenesis, or protection from apoptosis.

Comment 7. The statement that “autophagy inhibitors suppressed quercetin action” is important. Was this tested directly in vivo? Cite original data and expand the implication.

Response 7. All data included in our manuscript includes solely the results of in vivo studies. We have now added this information to the Introduction, page 3, lines: 104-107:

“This review summarizes results of in vivo rodent studies, which provide evidence that quercetin effectively mitigates key diabetes-related abnormalities, including hormonal and metabolic disturbances, oxidative and inflammatory stress, and alterations in the expression/action of the relevant enzymes and signaling molecules.”

The effects of quercetin on autophagy are indeed of particular interest and were investigated by Khater et al. (2024). In the section discussing the effects of quercetin on the pancreas (page 8, lines: 272-275) the following information has been added:

“In that study, rats were treated with 3-methyladenine (3-MA), an autophagy inhibitor that blocks autophagosome formation by inhibiting type III phosphatidylinositol 3-kinases (PI3K). 3-MA is a membrane-permeable compound commonly used to inves-tigate autophagy mechanisms [83].”

Comment 8. Hepatic IRS-1 and Akt pathway restoration is crucial, but the mechanistic discussion is very poor. Was this post-transcriptional modulation or direct inhibition of serine phosphorylation?

Response 8. We appreciate this insightful comment. Each article cited in our manuscript was carefully read and reviewed. While the observed restoration of hepatic IRS-1 and Akt signaling is indeed a key finding, the original studies unfortunately did not provide mechanistic detail regarding whether these effects were due to transcriptional regulation, post-transcriptional modulation, or inhibition of serine phosphorylation. We have now clarified this limitation in the manuscript by adding the following sentence, page 11, lines 371-373:

“However, the cited studies did not specify whether these effects involved transcrip-tional regulation, post-transcriptional modulation, or direct inhibition of serine phos-phorylation. Further investigation is needed to clarify the underlying mechanism.”

Comment 9. Throughout the manuscript, the English usage is overly repetitive (“quercetin therapy,” “diabetes-related disturbances”) and sometimes redundant. A careful language edit is recommended to improve clarity and flow.

Response 9. We fully acknowledge the reviewer’s concern regarding the repetitiveness and redundancy in the original manuscript. A comprehensive language revision has been undertaken to improve clarity, eliminate excessive repetition, and ensure better narrative flow.

Specifically, we reduced the frequency of recurring expressions by rephrasing or substituting them with more varied or context-specific. Where appropriate, we also removed redundant modifiers and streamlined sentence structures to improve readability without compromising scientific precision. Furthermore, the overall tone was adjusted to maintain a professional and concise academic style.

We hope that the revised version now meets the standards of linguistic accuracy and stylistic coherence expected by the journal.

Round 2

Reviewer 2 Report

Comments and Suggestions for Authors

The authors have revised the manuscript and addressed all of my comments and concerns. I recommend accepting the paper in its present form.